# Method Validation, Residues and Dietary Risk Assessment for Procymidone in Green Onion and Garlic Plant

**DOI:** 10.3390/foods11131856

**Published:** 2022-06-23

**Authors:** Li Li, Tingting Zhao, Yu Liu, Hongwu Liang, Kaiwei Shi

**Affiliations:** 1College of Plant Protection, Shanxi Agricultural University, Taiyuan 030031, China; 2School of Ecology and Environment, Inner Mongolia University, Hohhot 010020, China; ztt_20201010@163.com (T.Z.); ly15354952720@163.com (Y.L.); lianghongwu111@imu.edu.cn (H.L.); 3Institute for Pesticide Control, Ministry of Agriculture and Rural Affairs, Beijing 100125, China; zuo_ditie@126.com

**Keywords:** procymidone, green onion, garlic plant, residues, risk assessment

## Abstract

Procymidone is used as a preventive and curative fungicide to control fungal growth on edible crops and ornamental plants. It is one of the most frequently used pesticides and has a high detection rate, but its residue behaviors remain unclear in green onion and garlic plants (including garlic, garlic chive, and serpent garlic). In this study, the dissipation and terminal residues of procymidone in four matrices were investigated, along with the validation of the method and risk assessment. The analytical method for the target compound was developed using gas chromatography-tandem mass spectrometry (GC-MS/MS), which was preceded by a Florisil cleanup. The linearities of this proposed method for investigating procymidone in green onion, garlic, garlic chive, and serpent garlic were satisfied in the range from 0.010 to 2.5 mg/L with R^2^ > 0.9985. At the same time, the limits of quantification in the four matrices were 0.020 mg/kg, and the fortified recoveries of procymidone ranged from 86% to 104%, with relative standard deviations of 0.92% to 13%. The dissipation of procymidone in green onion and garlic chive followed first-order kinetics, while the half-lives were less than 8.35 days and 5.73 days, respectively. The terminal residue levels in garlic chive were much higher than those in green onion and serpent garlic because of morphological characteristics. The risk quotients of different Chinese consumer groups to procymidone in green onion, garlic chive, and serpent garlic were in the range from 5.79% to 25.07%, which is comparably acceptable. These data could provide valuable information on safe and reasonable use of procymidone in its increasing applications.

## 1. Introduction

Green onion (*Allium fistulosum* L.) and garlic (*Allium sativum* L.) are popular spices and vegetables worldwide with long cultivation histories, and both of them possess a variety of biological activities [1,2,3]. Green onions have antioxidant, anti-inflammatory, antifungal, and antibacterial effects [4,5,6] and are grown in almost all provinces in China, with a planting area of 550,000 square kilometers [6], accounting for approximately 3% of the total vegetables [2]. Welsh onion (*Allium fistulosum* L. var. *giganteum* Makino) and chive (*Allium schoenoprasum* L.) are the main varieties of green onion cultivated in northern and southern areas in China, respectively (Figure 1). Garlic and its preparations possess many health benefits according to epidemiological, experimental, and clinical evidence, such as lowering blood lipids and blood pressure and inhibiting the growth of microorganisms (viral, fungal, and bacterial) [7,8]. There are three edible portions in the growth stages of the garlic plant, including garlic (*Allium sativum* L.), garlic chive (*Allium sativum* L. var. *sativum*), and serpent garlic (*Allium sativum* L. var. *ophioscorodon* (Link) Döll) [9] (Figure 1). These edible garlic parts have their unique functions and flavors; consequently, planting area and output have been increasing in recent years [10].

Procymidone is a systematic dicarboximide fungicide [11], and it is used as a postharvest fungicide and seed treatment to control fungal growth on various crops [11,12,13]. It is used in vegetables (tomato, cucumber, and leek) and fruits (grape and melon) for the control of grey mold (*Botrytis cinerea*). The dissipation of procymidone in strawberry [14], tomato [15], cucumber [16], grape juice [17], leek [18], rape plant [19], and soil [20] has been studied, and the results indicated that procymidone has much longer half-lives in those matrices. More recently, procymidone residues were found to be always detectable in various crops, while the detection rates in pepper, leafy vegetables, and eggplant in Jinhua, Zhejiang Province, were 26.7%, 26.2%, and 13.6%, respectively [21]. Furthermore, procymidone has been proved to have potential toxicity toward non-target organisms in the environment and it could inhibit the growth of *Lemna minor* and *Scenedesmus Acutus* in aquatic plants [22], while showing more significant toxicity toward rainbow trout [23].

Procymidone can effectively control the gray mold which is usually found in the growth and storage periods of green onion, garlic, garlic chive, and serpent garlic. The maximum residue limit (MRL) for procymidone in Welsh onion is 2 mg/kg in Japan and 5 mg/kg in Korea, while in garlic it is 0.1 mg/kg and 5 mg/kg in Japan and Australia, respectively. MRLs for procymidone in garlic, green onion, garlic chive, and serpent garlic were established as 2 mg/kg, 7 mg/kg, 5 mg/kg, and 3 mg/kg in China, respectively [24]. However, the dissipation behavior and dietary risk of procymidone in green onion and garlic plants are not well-understood. Therefore, it is essential to obtain data on the dissipation and accumulation of procymidone in these vegetables and investigate the analytical method and risk assessment.

This study aimed to (1) optimize analytical methods using gas chromatography-tandem mass spectrometry (GC-MS/MS); (2) conduct field experiments in different areas across China to investigate procymidone’s dissipation rates and residue levels in four matrices; (3) assess the long-term dietary risks of procymidone based on the terminal residue results obtained. This study could provide a valuable reference for the safe and reasonable use of procymidone in green onion, garlic, garlic chive, and serpent garlic.

## 2. Materials and Methods

### 2.1. Chemicals and Reagents

Reference standards of procymidone (purity 98.0%) were obtained from Dr. Ehrenstorfer Gmbh, and 50% procymidone wettable powder was purchased from Hebei Zhongtianbangzheng Biologic Science Co., Ltd. (Cangzhou, Hebei, China). HPLC-grade acetonitrile was purchased from Merck (Darmstadt, Germany) and n-hexane from Thermo Fisher Technology Co., Ltd. (Shanghai, China). Sodium chloride, acetone, and petroleum ether of analytical grade were purchased from Beijing Chemical Reagent Co., Ltd. (Beijing, China). Petroleum ether was distilled before use and collected at 60 °C to 90 °C fractions. Florisil columns (1 g/6 mL) were purchased from ANPEL Laboratory Technologies, Inc. (Shanghai, China).

### 2.2. Stock Solution Preparation

Approximately 1000 mg/L of procymidone stock solution was prepared in acetone. The calibration and fortification standard solutions were freshly diluted with the stock solution using distilled petroleum ether/acetone (9:1, *v/v*). The matrix-matched standard solutions were prepared by diluting the stock solution with blank extracts of the matrices. The stock solution was stored at 4 °C in the dark.

### 2.3. Field Trials

The field trials were designed according to the Guideline on Pesticide Residue Trials (NY/T 788-2004) published previously by the Ministry of Agriculture and Rural Affairs, P.R. China, in order to obtain more residue data and complete evaluation of the behaviors of procymidone in green onion and garlic plants. In terms of geology and climate difference, representative regions for garlic and green onion were selected across the whole country. Field trials for the green onion were conducted in Shunyi District, Beijing (40.13N, 116.65E), Suzhou, Anhui (33.63N, 116.98E), Wulanchabu, Inner Mongolia (41.03N, 113.11E), Nanning, Guangxi (22.84N, 108.30E), Jinan, Shandong (36.82N, 117.21E), and Pengzhou, Sichuan (30.99N, 103.94E). The garlic field trials were undertaken in Daxing District, Beijing (39.73N, 116.34E), Suzhou, Anhui (33.63N, 116.98E), Zhengzhou, Henan (34.35N, 113.57E), Liuyang, Hunan (28.14N, 113.63E), Jinan, Shandong (36.82N, 117.21E), and Xuanwei, Yunnan (26.23N, 104.09E), in the years 2017 and 2018. Green onions and garlic plants in sets of six plots were cultivated in fields without a history of procymidone or other analogical compounds in the past year.

The varieties of green onion cultivated were Gaojiaobai in Beijing, Xincong No. 2 in Anhui, Chuncong No. 1 in Inner Mongolia, Xiangcong in Guangxi, Zheyindacong in Shandong, and Xiaocong in Sichuan Province. The varieties of garlic were Jinyu No. 3 in Beijing, Jinxiangbaisuan in Anhui and Shandong, Hongpidasuan in Hunan, Hebeidaming in Henan, and Hongqixing in Yunnan.

#### 2.3.1. Dissipation Trials

The dissipation trials were conducted for green onion and garlic chive at two sites each: Shunyi District, Beijing (40.13N, 116.65E), and Pengzhou, Sichuan (30.99N, 103.94E), for green onion; and Daxing District, Beijing (39.73N, 116.34E), and Suzhou, Anhui (33.63N, 116.98E), for garlic chive. In order to avoid random errors, the trials were conducted in triplicate, and one other plot was set as a control. Each plot was at least 15 m^2^ and separated using a buffer area of one meter from the following plot. A 50% procymidone wettable powder was used to obtain the dissipation trends in green onion and garlic chive. It was applied at a 675 g a.i./ha by spraying once during the early stage of the disease.

#### 2.3.2. Terminal Residue Trials

The investigations of the terminal residues of procymidone and evaluation of their safe use in green onion, garlic, garlic chive, and serpent garlic were conducted in the experimental fields. The terminal residue study used two different 50% procymidone wettable powder dosages: one was the recommended dosage with a rate of 450 g a.i./ha and the other was 1.5 times the recommended dosage with a rate of 675 g a.i./ha. Two dosages were applied with a 7-day interval two or three times. The maturing stages of the garlic, garlic chive, and serpent garlic were different. Consequently, three separate treatment areas were set. As in the dissipation trial, each treatment plot consisted of 15 m^2^ with three replicates. There were buffer areas to avoid cross-contamination between the different treatments.

#### 2.3.3. Sample Collection and Preparation

For the dissipation study, green onion and garlic chive samples were randomly collected at 0 (2 h after application), 1, 3, 5, 7, 10, 14, 21, and 30 days after application.

For the terminal residue study, garlic chive and serpent garlic samples were randomly collected 5, 7, and 10 days, and garlic samples were collected 7, 14, and 21 days after the last application at harvest time. In contrast, green onion samples were collected 5, 7, 10, and 14 days after the last application.

The corresponding blank samples of the four matrices were collected. The blank and treated samples were cut into small pieces, crushed with a Foss HM294 homogenizer at a speed of 1500 rpm, and kept at −20 °C until analysis to avoid any degradation of residues.

### 2.4. Sample Extraction and Purification

The crushed samples (10.00 ± 0.01 g) of green onion, garlic, garlic chive, or serpent garlic were weighed into a 50 mL centrifuge tube and extracted with 20 mL of acetonitrile using ultrasonication for 15 min. Then, 6 g of sodium chloride was added, and the tube was shaken vigorously for 1 min and centrifuged at 3000 rpm for 5 min. An aliquot of 10 mL supernatant was evaporated to near dryness using a vacuum rotary evaporator at 35 °C. The Florisil SPE cartridge had been previously conditioned with 5 mL of petroleum ether, and the eluent was dropped. The concentrated extracts were washed with 5 mL of petroleum ether/acetone (9:1, *v/v*) and transferred to the cartridge twice. Subsequently, the eluent was collected and concentrated to dryness on a vacuum rotary evaporator at 35 °C and dissolved in 2.5 mL of petroleum ether/acetone (9:1, *v/v*) for GC-MS/MS analysis.

### 2.5. GC-MS/MS Analysis

Procymidone residue was detected using a 7890 GC equipped with a 7000 C Triple Quadrupole mass spectrometer (Agilent Technologies, Santa Clara, CA, USA). The target compound was chromatographically separated with a DB-1701 column (30 m × 0.25 mm i.d. × 0.25 µm film thickness). The interface, ion source, and quadrupole temperatures were set at 280 °C, 300 °C, and 150 °C, respectively. The injection volume was 2 μL in splitless mode at 280 °C. Separation was performed using helium at a 1 mL/min constant flow rate. The oven temperature was programmed as follows: the initial temperature was 150 °C, which was raised to 255 °C at 6 °C/min, held for 1 min, then raised to 270 °C at 25 °C/min and held for 1 min. Under the conditions mentioned above, the quantitative and qualitative ions were m/z 282.7/95.9 and 282.7/254.8, and the retention time of procymidone was 15.53 min.

### 2.6. Storage Stability

To evaluate the concentration variation in frozen green onion, garlic, garlic chive, and serpent garlic samples, a storage stability study was further conducted with the sample collection. An aliquot of 0.1 mL of 10 mg/L working standard solutions was added to 10 g crushed blank samples, resulting in a final concentration of 0.1 mg/kg. Green onion samples were determined at intervals of 2, 4, and 9 months, while garlic, garlic chive, and serpent garlic samples were determined at intervals of 1, 3, and 11 months.

### 2.7. Data Analysis and Calculation

The matrix effect (ME) was assessed by comparing the slopes of the matrix-matched standard calibration with the slope in MeCN. A higher slope for the matrix standard indicated matrix-induced signal enhancement, whereas a lower slope indicated signal suppression [25]. The ME could be categorized as soft MEs (0 < |ME| ≤ 20%), medium MEs (20% < |ME| < 50%), and strong MEs (|ME| ≥ 50%). The soft MEs could be ignored. The following equation was used to evaluate the ME:(1)$$\mathrm{ME},\%=\left(\frac{\mathrm{the}\;\mathrm{slope}\;\mathrm{of}\;\mathrm{matrix}\;\mathrm{standard}}{\;\mathrm{the}\;\mathrm{slope}\;\mathrm{of}\;\mathrm{solvent}\;\mathrm{standard}}-1\right)\times 100\%$$


The dissipation dynamics and half-lives of procymidone in green onion, garlic, garlic chive, and serpent garlic were determined using first-order kinetics equations:(2)$$C_{t}=C_{0}e^{-\mathrm{kt}}$$


The half-lives (t_1/2_) were calculated using the equation:(3)$$t_{1/2}=\frac{\ln2}{k}$$
where C_t_ (mg/kg) was the fungicide residue concentration at time t, C_0_ (mg/kg) was the initial concentration, and k was the first-order rate constant, which was the degradation rate (day^−1^) [14].

The dietary risk assessment was conducted based on the terminal residue data for the green onion, garlic, garlic chive, and serpent garlic obtained from field trials. National estimated daily intake (NEDI) and risk quotients (RQs) were used to evaluate the dietary risk of procymidone in four matrices. The following equations were used to evaluate the NEDI and RQs:(4)$$\mathrm{NEDI}=\frac{\sum \left(\mathrm{STMRi}\times \mathrm{Fi}\right)}{\mathrm{bw}}$$
(5)$$\mathrm{RQ}=\frac{\mathrm{NEDI}}{\mathrm{ADI}}\times 100\%$$
where STMRi (mg/kg) was the supervised trial median residue, Fi (kg) was the dietary reference intake, bw (kg) was the body weight, and ADI (mg/kg BW) was the acceptable daily intake of pesticide [25]. Any RQ value higher than 100% indicates an unacceptable human risk from the pesticide [26].

The statistical software package SPSS 20.0 (IBM SPSS Software, Chicago, IL, USA) was used to conduct the statistical analysis. A value of *p* < 0.05 was considered to be statistically significant.

## 3. Results and Discussion

### 3.1. Analytical Methods

#### 3.1.1. Selection of Purification Method

There is an existing and effective national standard in China to determine procymidone in various vegetables and fruits (NY761-2008) [27]. Procymidone residues in samples were extracted with acetonitrile, cleaned up with a Florisil column, and detected with a gas chromatography-electron capture detector (GC-ECD). In this study, an analytical method was also attempted based on an NH_2_ column for cleanup and toluene/acetonitrile (1:3, *v/v*) as the eluent for garlic samples. The chromatography of the garlic extracts using these two methods is shown in Figure 2, but it was impossible to quantify samples in this condition because of the increased interference.

Many sulfur-containing compounds exist in green onion and garlic [28], and 3, 4-dimethyl thiophene, dipropyl disulfide and methyl propyl disulfide have been identified in green onion [29]. These compounds would interfere with the quantitative and qualitative analysis under GC conditions. Treating samples from similar crops, such as onion, leek, and ginger, with microwaves resulted in effectively inactivated enzymes and matrix effects were eliminated [30,31]. Consequently, a microwave method using middle power for 30, 40, or 50 min was attempted; the impurities decreased with heating time, and the recovery of procymidone also decreased from 62% to 19%. Therefore, the analytical methods based on the Florisil column, NH_2_ column, and microwave could not meet the quantitation requirements for procymidone residue analysis. Finally, an analytical method based on MeCN extraction, Florisil cleanup, and GC-MS/MS detection was adopted to analyze the residues of procymidone in four matrices.

#### 3.1.2. Method Validation

Method validation was performed by checking the calibration curve, matrix effect (ME), limit of quantification (LOQ), accuracy, and precision.

The linearity was constructed by plotting concentrations against the peak area, and appropriate linearity and coefficients over reasonable concentration ranges were achieved. All regression data for procymidone in the solvent, green onion, garlic, garlic chive, and serpent garlic were obtained, as shown in Table 1. Procymidone exhibited good linearity over a concentration of 0.010–2.5 mg/L in the four matrices, and the correlation coefficients were higher than 0.9985. The MEs of procymidone in garlic chive and serpent garlic were −13.48% and −18.52%, which indicated soft matrix effects and could be ignored. Medium matrix effects in the green onion and garlic were observed with values of 38.00% and −21.30%. Therefore, matrix-matched calibration standards were used for quantification to ensure the method’s accuracy. The LOQ was defined as the minimum fortified level of the target compound, which was 0.02 mg/kg in the four matrices.

The precision and accuracy of the method were evaluated by calculating fortified recoveries at three or four different fortified levels with five replications (*n* = 5). As shown in Table 2, the mean recoveries of procymidone in green onion were 92% to 104% with relative standard deviations (RSDs) of 3.3% to 5.8% at fortified levels of 0.020, 0.050, 0.50, 5.0, and 10 mg/kg, respectively. The mean recoveries of procymidone in garlic, garlic chive, and serpent garlic were 86% to 98%, 84% to 102%, and 85% to 97% with relative standard deviations (RSDs) of 0.92% to 13%, 2.8% to 4.5%, and 5.3% to 8.6% at spiked levels of 0.020, 0.50, 5.0, and 60 mg/kg (only for garlic chive), respectively. Typical GC-MS/MS chromatograms of procymidone are shown in Figure 3. The above findings demonstrated that the proposed method was suitable and adequate for procymidone residues in green onion, garlic, garlic chive, and serpent garlic samples.

### 3.2. Storage Stability of Procymidone in Green Onion, Garlic, Garlic Chive, and Serpent Garlic

As shown in Table 3, the residues of procymidone changed within certain ranges during the storage period at −20 °C in the dark, ranging from 0.093 to 0.099 mg/kg in green onion, 0.081 to 0.087 mg/kg in garlic, 0.072 to 0.093 mg/kg in garlic chive, and 0.082 to 0.091 mg/kg in serpent garlic. The mean degradation rates for the green onion, garlic, garlic chive, and serpent garlic ranged from −1.0% to 5.1%, 2.8% to 7.0%, −1.6% to 22.1%, and −1.9% to 8.1%, respectively. The results showed that the degradation rates were lower than 30%, which indicated that procymidone remained comparably stable during the period from sample collection to testing [32].

Among the four matrices, the degradation rate in garlic chive was much higher than that in garlic and serpent garlic over the same period. According to Bian et al. [33] and Sun et al. [34], the storage temperature, moisture content, pH, and enzyme activity of the matrices should be the main influencing factors for the storage stability of compounds. The specific mechanisms explaining the variance in procymidone storage stability in the different matrices require further verification.

### 3.3. The Dissipation Pattern of Procymidone in Green Onion and Garlic Chive

Following the relevant literature, the initial residues were defined as the residue levels in the samples 2 h after pesticide application [35]. In this study, the dosages of 50% procymidone wettable powder in green onion and garlic chive were both 675 g a.i./ha. The initial concentrations in green onion were 2.65 mg/kg (Beijing) and 5.80 mg/kg (Sichuan), and they were 10.49 mg/kg (Beijing) and 9.85 mg/kg (Anhui) in garlic chive. The initial concentrations in garlic chive were significantly higher than those in green onion, which was perhaps caused by the crops’ morphological characteristics. The garlic chive was leafy with a fluffy surface and the whole leaf was flat, allowing it to carry more droplets, while the green onion was cylindrical. The initial concentration in the Sichuan samples was approximately twice that of the Beijing samples, which was perhaps caused by the different crop varieties. The species in Beijing was Welsh onion (*Allium fistulosum* L. var. *giganteum* Makino), and a part of the plant was under the soil. In Sichuan, chive (*Allium fistulosum* L. var. *caespitosum* Makino) was cultivated, and most of the plant was above the ground and contained many more sprayed droplets.

The main dissipation-related parameters—namely, the regression equations, correlation coefficients, half-lives, and degradation trends for procymidone in green onion and garlic chive—are shown in Table 4 and Figure 4. The residues of procymidone in green onion and garlic chive dissipated following first-order kinetics, with a regression coefficient of the degradation equation of 0.9282 to 0.9857. The half-lives of green onion in Beijing and Sichuan and garlic chive in Beijing and Anhui were 6.93, 8.35, 5.78, and 4.33 days, respectively. Although the garlic chive’s initial concentration was higher than that of the green onion, the half-life was shorter. In previous studies, the half-life of procymidone in strawberries has been reported to be 8.7 to 12.2 days [14], while that in green bean plantations was 10.9 ± 1.4 days [36] and that in ginseng plants was 18.28 days [37]. In general, the half-lives of procymidone residues in garlic chive were shorter than those in studies on various other crops. The variability in the dynamics kinetics can be affected by factors such as the physicochemical properties of pesticides, the application doses, the application method, the crop variety, the sampling tissues, the local environment, and crop growth characteristics [38,39], all of which can play comprehensive roles in the dissipation process in pesticides in crops. This study considered the crop variety, the local environment, and crop growth characteristics as the main factors affecting the green onion and garlic chive dynamics kinetics.

### 3.4. Terminal Residues of Procymidone in Green Onion, Garlic, Garlic Chive, and Serpent Garlic

Two doses (450 g a.i./ha and 675 g a.i./ha) of procymidone, sprayed twice or three times, were applied to evaluate the terminal residue levels in green onion garlic, chive, and serpent garlic. The terminal residues of procymidone for different spraying times and dosages in green onion at six sites are shown in Figure 5. Residues in Anhui, Beijing, Inner Mongolia, and Shandong were significantly lower than those in Guangxi and Sichuan. The different varieties of green onion may have been the cause of this phenomenon.

Terminal residues for low and high dosages sprayed twice and three times in serpent garlic and garlic chive are shown in Figure 6 and Figure 7. In the results for garlic chive, residues in the Hunan plant were found to be significantly higher than residues at other locations. The climate, rainfall, and temperature may have been the main factors, and plant height and density also influenced the residues. Furthermore, as shown in Figure 8, in the three stages of garlic in Beijing, the terminal residues in garlic chive were the highest, while residues in garlic were below the LOQ. It is likely that the aboveground part contained higher procymidone residues than the underground part. The surface of the serpent garlic was smooth and not easy to cover with pesticides [40] compared to the serpent garlic due to morphological characteristics, and the results were consistent with Bian’s study [10].

### 3.5. Dietary Exposure Risk Assessment

Public attention to food safety has been increasingly significantly in recent times, and food consumption is the main route of human exposure to environmental pollutants [41]. In this part of the study, chronic dietary risk was evaluated based on dietary consumption and body weight (bw) in accordance with the Chinese National Nutrition and Health Survey program [42]. The survey was undertaken in a big city, medium and small cities, and rural areas, and the rural areas were classified as I~IV according to the regions and their overall economic strength. The STMRi values for the dietary risk assessment in green onion, garlic chive, and serpent garlic were 0.70 mg/kg, 3.98 mg/kg, and 0.64 mg/kg, respectively, and the ADI value for procymidone was 0.1 mg/kg bw [24]. As shown in Figure 9, the total risk quotient for the age 14–70 consumer group was relatively low. The age 2 to 13 group suffered the higher chronic exposure to procymidone. The risk for IV rural females aged 2 to 3 with greater vegetable consumption and lower weight was highest, and the maximum RQ reached 25.07%. These results indicate that procymidone’s total chronic dietary risk in the four matrices for Chinese residents in different consumer groups was acceptable. These results provide essential data for determining the risks associated with green onion, garlic chive, serpent garlic, and garlic consumption.

## 4. Conclusions

An analytical method based on GC-MS/MS was validated for procymidone in green onion, garlic, garlic chive, and serpent garlic. Our dissipation study showed that the half-lives of procymidone were 6.86 days and 8.35 days in green onion in Beijing and Sichuan, and they were 5.73 days and 4.36 days in garlic chive in Beijing and Anhui. The residue levels of procymidone were significantly influenced by crop morphology, resulting in the terminal residue levels in garlic chive being much higher than those in the other three matrices. Based on the safety risk assessment using residue data, the RQs of different Chinese consumer groups were between 5.79% and 25.07%, indicating a low and acceptable dietary exposure risk for procymidone in green onion, garlic chive, and serpent garlic. This study can provide field data for the dietary risk assessment of procymidone in green onion, garlic, garlic chive, and serpent garlic.

## Figures and Tables

**Figure 1 foods-11-01856-f001:**
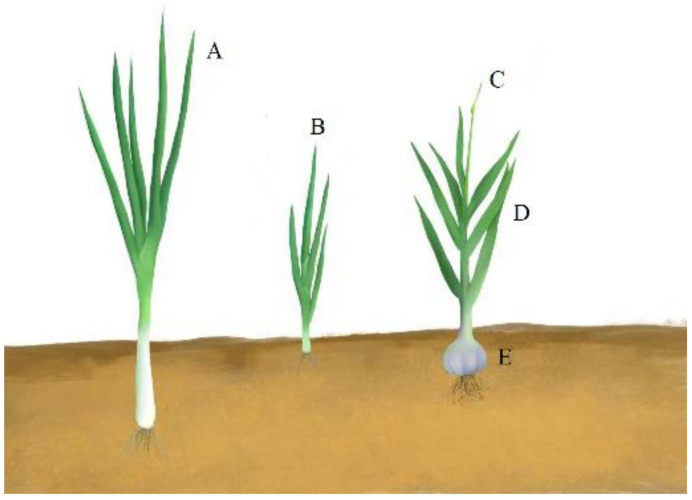
Schematic diagram of different varieties of green onion and parts of the garlic plant. (**A**) Welsh onion (*Allium fistulosum* L. var. *giganteum Makino*); (**B**) chive (*Allium schoenoprasum* L.); (**C**) serpent garlic (*Allium sativum* L. var. *ophioscorodon* (Link) Döll); (**D**) garlic chive (*Allium sativum* L. var. *sativum*); (**E**) garlic (*Allium sativum* L.).

**Figure 2 foods-11-01856-f002:**
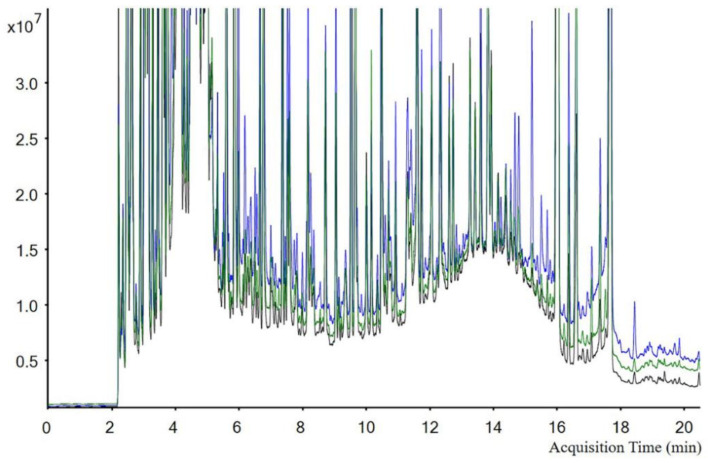
Chromatogram of different cleanup conditions. Black line: blank garlic sample; blue line: garlic sample purified with NH_2_ column; green line: garlic sample purified with Florisil column.

**Figure 3 foods-11-01856-f003:**
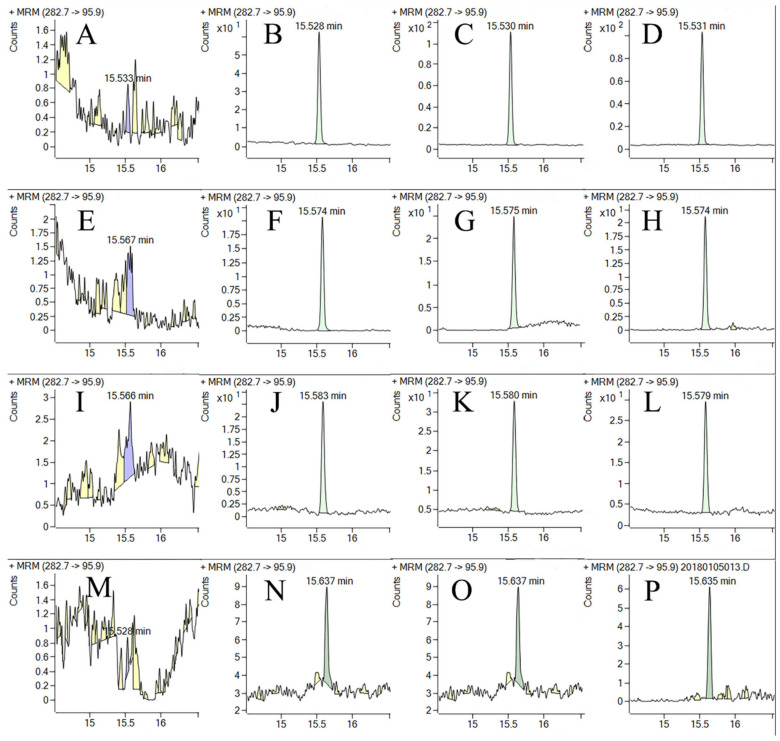
Typical chromatograms of procymidone in green onion (**A**–**D**), garlic chive (**E**–**H**), serpent garlic (**I**–**L**), and garlic (**M**–**P**). (**A**,**E**,**I**,**M**) are for the blank sample; (**B**,**F**,**J**,**N**) are for the solvent standard at 0.04 mg/kg; (**C,G,K,O**) are for the matrix standard at 0.04 mg/kg; and (**D**,**H**,**L**,**P**) are for the spiked sample at 0.02 mg/kg.

**Figure 4 foods-11-01856-f004:**
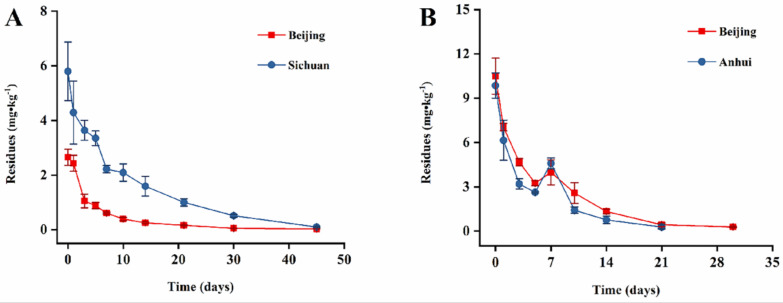
The dissipation of procymidone at different experiment sites. (**A**) Green onion; (**B**) garlic chive.

**Figure 5 foods-11-01856-f005:**
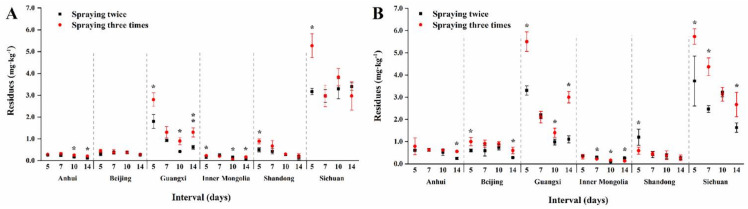
Terminal residues of procymidone for different spraying times in green onion. The asterisks represent statistically significant differences between two spraying times on the same interval day (*p* < 0.05). (**A**) low dosage; (**B**) high dosage.

**Figure 6 foods-11-01856-f006:**
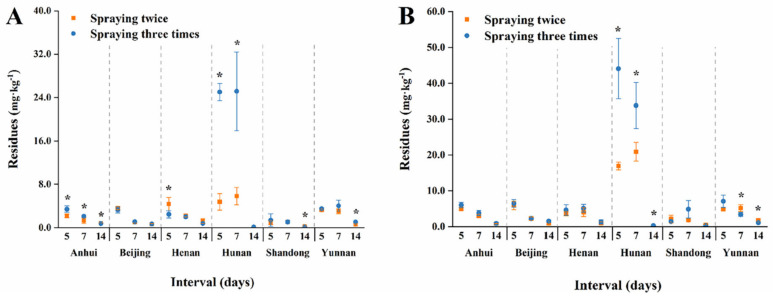
Terminal residues of procymidone for different spraying times in garlic chive. The asterisks represent statistically significant differences between two spraying times on the same interval day (*p* < 0.05). (**A**) low dosage; (**B**) high dosage.

**Figure 7 foods-11-01856-f007:**
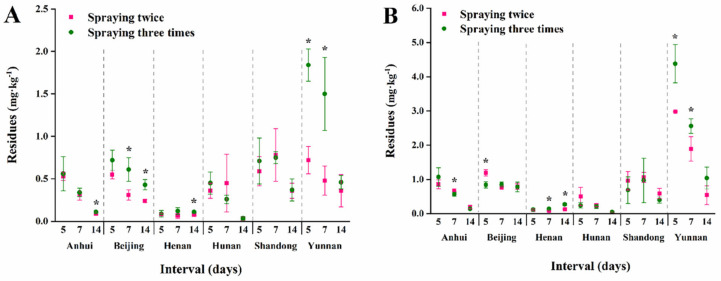
Terminal residues of procymidone for different spraying times in serpent garlic. The asterisks represent statistically significant differences between two spraying times on the same interval day (*p* < 0.05). (**A**) low dosage; (**B**) high dosage.

**Figure 8 foods-11-01856-f008:**
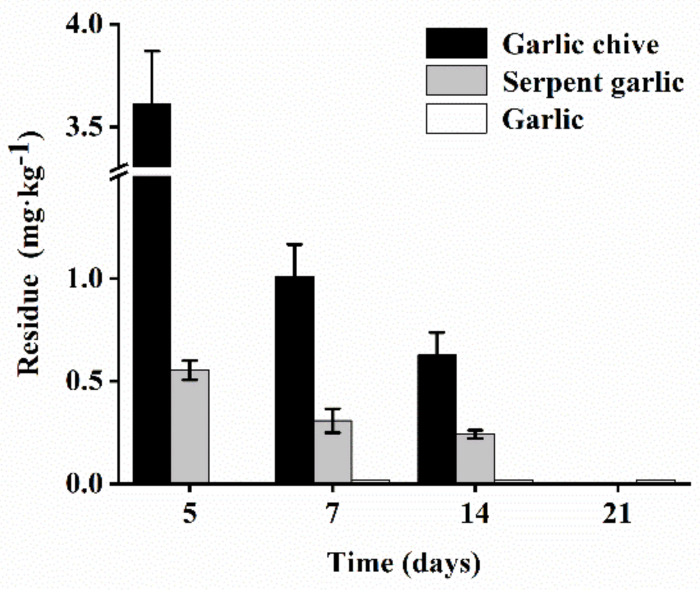
Terminal residues of procymidone in garlic chive, serpent garlic, and garlic in Beijing.

**Figure 9 foods-11-01856-f009:**
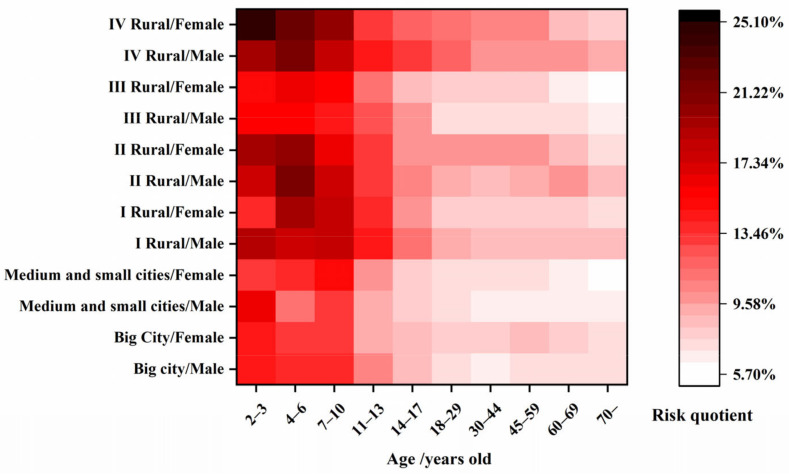
Chronic dietary risk for different consumer groups at different ages.

**Table 1 foods-11-01856-t001:** Linearity ranges, regression equations, matrix effects, and limits of quantification for procymidone in a solvent, green onion, garlic, garlic chive, and serpent garlic.

Matrix	Linearity Range (mg·L^−1^)	Regression Equation	R^2^	Matrix Effect (ME, %)	Limit of Quantification (LOQ, mg·kg^−1^)
Solvent	0.010–2.5	*y* = 3582.5*x* − 74.202	0.9985	/	/
Green onion	0.010–2.5	*y* = 4944.1*x* + 92.228	0.9998	38.00	0.020
Garlic	0.010–2.5	*y* = 2819.6*x* + 48.745	0.9996	−21.30	0.020
Garlic chive	0.010–2.5	*y* = 3099.6*x* + 213.560	0.9998	−13.48	0.020
Serpent garlic	0.010–2.5	*y* = 2919.2*x* + 64.964	0.9998	−18.52	0.020

**Table 2 foods-11-01856-t002:** The mean recoveries and RSDs of procymidone in four matrices at different fortified levels (*n* = 5).

Matrix	Fortified Levels (mg·kg^−1^)	Mean Recoveries (%)	RSDs (%)
Garlic chive	0.020	92	4.5
	0.50	100	3.3
	5.0	102	3.6
	60	84	2.8
Serpent garlic	0.020	97	8.6
	0.50	95	5.3
	5.0	85	7.2
Garlic	0.020	95	13
	0.50	98	0.92
	5.0	86	3.1
Green onion	0.020	104	5.8
	0.050	92	4.9
	0.50	92	5.1
	5.0	95	3.7
	10	93	3.3

**Table 3 foods-11-01856-t003:** Residues and degradation rates of procymidone in green onion, garlic, garlic chive, and serpent garlic matrices during laboratory storage.

Time (Months)	Green Onion		Time (Months)	Garlic		Garlic Chive		Serpent Garlic	
	R ^a^ (mg·kg^−1^)	DR ^b^ (%)		R (mg·kg^−1^)	DR (%)	R (mg·kg^−1^)	DR (%)	R (mg·kg^−1^)	DR (%)
0	0.098	-	0	0.087	-	0.092	-	0.089	-
1	0.099	−1.0	2	0.083	4.2	0.093	−1.6	0.091	−1.9
3	0.098	0.0	4	0.085	2.8	0.089	3.5	0.083	6.8
11	0.093	5.1	9	0.081	7.0	0.072	22.1	0.082	8.1

^a^ R: residue. ^b^ DR: degradation rate.

**Table 4 foods-11-01856-t004:** The dosage, experiment sites, regression equations, correlation coefficients, and half-lives (t_1/2_) of procymidone in green onion and garlic chive.

Matrix	Dosage (g a.i·ha ^−1^)	Experiment Sites	Regression Equation (*y*=)	Correlation Coefficient (*r*)	Half-Life (Days)
Green onion	675	Beijing	1.53e^−0.10*x*^	0.9282	6.93
		Sichuan	5.00e^−0.08*x*^	0.9857	8.35
Garlic chive	675	Beijing	7.79e^−0.12*x*^	0.9608	5.78
		Anhui	7.64e^−0.16*x*^	0.9293	4.33

## Data Availability

Data are contained within the article.

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
