# Peer review of "Method Validation, Residues and Dietary Risk Assessment for Procymidone in Green Onion and Garlic Plant"

_foods, 2022, doi:10.3390/foods11131856_

Round 1

Reviewer 1 Report

The main goal of agriculture is to produce safe and high-quality food for an ever-growing population of people around the world. With the development of techniques to increase production efficiency, the problems associated with the use of pesticides in agriculture are also increasing First of all, it is necessary to consider the toxic impact on human health and the environment. Consequently, it is essential to evaluate dissipation, accumulation, and risk assessment of pesticides in various fruits and vegetables. In this study the authors optimized the analytical method for determination of procymidone in four matrices: green onion, garlic, garlic chive, and serpent garlic, using gas chromatography coupled with tandem mass spectrometry (GC-MS/MS). They are also investigated procymidone's dissipation rates of these matrices. Moreover, authors assess the long-term dietary risks of procymidone based on obtained terminal residue levels. According to the reviewer this Article is well prepared, method is well described, text contain essential parts, the results and discussion part is well prepared and summarized with relevant conclusions. Reviewer has minor comments.

According to the reviewer this Article is well prepared. The reviewer has some suggestions for the text:

1.       In the Abstract: The analytical method of the target compound was developed using gas chromatography-tandem mass spectrometry (GC-MS/MS) coupled with Florisil cleanup. Instead of coupled it would be better use preceded by.

2.       In material and method section, authors described Field trials and it should be clearly stated in the text whereTerminal residue trials’ and ‘Dissipation trials’ were carried out.

3.       The linearity was set at a range of 0.010-2.5 mg/L and the fortification were performed at the highest concentration level of 5 mg/kg. Why?

Author Response

       Thank you for your review and suggestions on the manuscript “Method validation, residues, and dietary risk assessment of procymidone in green onion and garlic plant (No. 1758944)”. The comments are valuable and helpful for the promotion of our manuscript. We have made corrections carefully, and revisions are shown using red highlight in the text. Responses are presented following.

  1. In the Abstract: The analytical method of the target compound was developed using gas chromatography-tandem mass spectrometry (GC-MS/MS) coupled with Florisil cleanup. Instead of coupled it would be better use preceded by.

Response: Thank you for your suggestions, and “coupled” had been revised to “preceded by” in the text.

  1. In material and method section, authors described Field trials and it should be clearly stated in the text where‘Terminal residue trials’ and ‘Dissipation trials’ were carried out.

Response: The sites for terminal residue trials and dissipation trials had been added in Line 96~102 and Line 107~110 in the text.

  1. The linearity was set at a range of 0.010-2.5 mg/L and the fortification were performed at the highest concentration level of 5 mg/kg. Why?

Response: To ensure the effectiveness of the analytical method, the fortified level should cover the highest residues. Consequently, the fortification was set at 5mg/kg in garlic and garlic chive, while 10mg/kg in garlic serpent and 60mg/kg in green onion,respectively. However, according to the response of the detector, the linearity can only cover the range of 0.010-2.5 mg/L, and concentrations higher than 2.5 mg/L had been diluted to the range of 0.01 and 2.5 mg/L.

Reviewer 2 Report

The manuscript concerns methodology for procymidone determination in vegetables and also assessment of consumer exposure to this pesticide. This active substance was very popular in European Union till 2008. Now it is not approved for application as pesticide in Europe. Therefore this manuscript could be of less interest for people from Europe.

The validation analysis and field trials were performed very precisely.

line 10 – The better word is “control” or “reduce” instead of “curb”

line 119 – There is “….each treatment consisted of 15m2”, it would be better “….each treatment plot consisted of 15m2……”

The phrase “Isolation belts” is not good. Better one is “buffer area”

Specify and add information how samples were crushed.

What is it “garlic ck”?

Line 196 - Why authors use gas chromatography-electron capture detector (GC-ECD) not GC-MS?

Line 251 “The influencing factors on the storage stability of compounds were closely related to the storage temperature, pH, and moisture content of matrices and the corresponding enzyme activity [33,34].” In my opinion authors should change this sentence to “According to Bian et al. and Sun et al factors influencing on the storage stability of…………………” – The first sentence suggest that authors of this manuscript performed such studies for procymidone.

Fig. 9 – Add information what is it I Rural, II Rural, III Rural and IV Rural. The same concerns results section (line 317).

Author Response

Thank you for your review and suggestions on the manuscript “Method validation, residues, and dietary risk assessment of procymidone in green onion and garlic plant (No. 1758944)”. The comments are valuable for us. We have made corrections carefully, and revisions are shown using red highlight in the text. Responses are as follows:

  1. line 10 – The better word is “control” or “reduce” instead of “curb”

Response: “curb” in Line 10 and Line 45 had been revised to “control” in the manuscript.

  1. line 119 – There is “….each treatment consisted of 15m2”, it would be better “….each treatment plot consisted of 15m2……”

Response: “plot” had been added in Line 127, and it was more accurate now.

  1. The phrase “Isolation belts” is not good. Better one is “buffer area”

Response: In Line 128, the “Isolation belts” had been revised to “buffer area”.

  1. Specify and add information how samples were crushed.

Response: The specific information “crushed with a homogenizer of Foss HM294 at a speed of 1500rpm” had been added in Line 137~138.

  1. What is it “garlic ck”?

Response: “garlic ck” in the caption of Fig.1 means “blank garlic sample”, and it had been revised this time.

  1. Line 196 - Why authors use gas chromatography-electron capture detector (GC-ECD) not GC-MS?

Response: According to the Guideline on Pesticide Residue Trials (NY/T 788-2004), procymidone residues in vegetables and fruits were analyzed by GC-ECD, consequently, the determination was first tried with GC-ECD.

  1. Line 251 “The influencing factors on the storage stability of compounds were closely related to the storage temperature, pH, and moisture content of matrices and the corresponding enzyme activity [33,34].” In my opinion authors should change this sentence to “According to Bian et al. and Sun et al factors influencing on the storage stability of…………………” – The first sentence suggest that authors of this manuscript performed such studies for procymidone.

Response: The sentence had been revised according to your suggestions in Line 260-261.

  1. 9 – Add information what is it I Rural, II Rural, III Rural and IV Rural. The same concerns results section (line 317).

Response: In our manuscript, the chronic dietary risk assessment was evaluated based on the dietary consumption and body weight(bw) according to the Chinese National Nutrition and Health Survey program (Survey Report on the Nutrition and Health of Chinese Residents: Data Set on the Status of Nutrition and Health in 2002). The survey was finished in big city, medium and small cities and rural, and the rural was divided as I~IV mainly according to the regions and overall economic strength. Relevant contents are reflected in line 325.

Reviewer 3 Report

The manuscript presents research on the validation of the procymidone determination method in selected vegetables from the Alliaceae family. In addition, the authors conducted field experiments to evaluate procymidone residues and determined the health risks associated with the presence of procymidone in the tested vegetables.

The presented topic is interesting. However, the work must be prepared in accordance with the journal's requirements (size of fonts, places of tables and figures, layout of the manuscript) - The authors did not use the editorial matrix of the manuscript (https://www.mdpi.com/journal/foods/instructions). The formulas for estimating dietary risk assessment are vaguely described - the numerical values on the basis of which the risk related to taking this pesticide with the diet were assessed should be clearly provided. What is the consumption of these weights in China? I do not see where the Authors presented the results on risk quotients (RQ)?

Author Response

Thank you very much for your comments and we tried our best to revise our manuscript according to your suggestions. The following are the responses and revisions I have made.

The presented topic is interesting. However, before publication, the work must be prepared in accordance with the journal's requirements (size of fonts, places of tables and figures, layout of the manuscript) - The authors did not use the editorial matrix of the manuscript (https://www.mdpi.com/journal/foods/instructions). The formulas for estimating dietary risk assessment are vaguely described - the numerical values on the basis of which the risk related to taking this pesticide with the diet were assessed should be clearly provided. What is the consumption of these weights in China? I do not see where the Authors presented the results on risk quotients (RQ)?

Response: Thank you again for your comments. Our manuscript had been revised in accordance with the journal's requirements in this version. It is really true that the dietary consumption and body weight(bw) in the formulas had not been provided in the manuscript. In this study, the chronic dietary risk assessment was evaluated based on the dietary consumption and body weight(bw) of different regions and ages according to the Chinese National Nutrition and Health Survey program(Survey Report on the Nutrition and Health of Chinese Residents: Data Set on the Status of Nutrition and Health in 2002). The corresponding data were much more and it would be confused in the form of table. Consequently, the risk quotients(RQ) were provided in Fig.9 by the heatmap, and I think it was more specific in this form.

Round 2

Reviewer 3 Report

The authors revised the manuscript. In the version presented now, the authors clearly stated the criterion for assessing the level of risk quotients - the data are expressed as a percentage and the criteria are given as a percentage. However, I would like to point out once again that the manuscript does not follow the template of the Foods journal (https://www.mdpi.com/journal/foods/instructions).